



# A simple improvement of a tip loss model for actuator disc and actuator line simulations

Georg Raimund Pirrung[1] and Maarten Paul van der Laan[1]

[1]Wind Energy Department, Technical University of Denmark, Frederiksborgvej 399, DK-4000 Roskilde, Denmark

**Correspondence:** Georg R. Pirrung (gepir@dtu.dk)

**Abstract.** The loading of a wind turbine decreases towards the blade tip due to the velocities induced by the tip vortex and due to the spanwise flow. This tip loss effect has to be taken into account when performing actuator disc simulations, where the single blades of the turbine are not modeled, and when performing actuator line simulations, where the resolution typically is not fine enough to properly resolve the tip vortex. A widely used method applies a factor on the axial and tangential loading of the turbine. This factor decreases when approaching the blade tip. It has been shown that the factor should be different for the axial and tangential loading of the turbine due to the rotation of the resulting force vector at the airfoil sections caused by the induced velocity. The present article contains the derivation of a simple correction for the tangential load factor that takes this rotation into account. The correction does not need any additional curve fitting but just depends on the local airfoil characteristics and angle of attack.

*Copyright statement.* TEXT

## 1 Introduction

Actuator disc (AD) (Mikkelsen, 2003) or actuator line methods (Sørensen and Shen, 2002) enable the studying of wind turbine wake effects in computational fluid dynamics (CFD) at a much lower computational effort than full rotor CFD. This reduced effort is mainly due to a coarser computational grid, which does not resolve the detailed flow around the individual blades. Instead, the aerodynamic coefficients are looked up from 2D airfoil polar tables based on the local flow velocity and angle of attack. The resulting forces are then applied on the flow in the CFD domain as permeable body forces (Réthoré et al., 2014). The reliance on airfoil data and the coarse resolution of the CFD grid makes a tip loss correction necessary. This is because two effects are not resolved numerically: 1) the induced velocity due to the tip vortex and 2) the changed pressure distribution around the airfoil due to the spanwise velocity. Wimshurst and Willden (2018) found that the second effect starts to develop from 95 % of the rotor radius, where the pressure distribution from 3D full rotor computations is different from 2D pressure distributions at any angle of attack.

The tip loss correction proposed by Shen et al. (2005), which has been widely adapted in the industry, consists of a scaling factor $F_1^{\text{Shen}}$ on the aerodynamic forces that approaches zero towards the blade tip. Wimshurst and Willden (2017) showed that





a better comparison to full rotor measurements could be obtained by fitting correction factors for the normal and the tangential forces independently. This means that the resulting force vector is not just scaled as in the original Shen et al. (2005) correction but also rotated.

The motivation for the present work is to introduce a new method of determining the tangential force, such that a separate
fitting of the tip loss factor for the tangential force can be avoided. This method neglects the effect of the increasing spanwise velocity towards the tip and instead is based on the assumption that the tip loss effect is dominated by the induced velocity due to the tip vortex. This is similar to how tip loss is treated in BEM implementations (Glauert, 1963) and how tip loss arises in lifting line vortex codes. The induced velocity causes a change in angle of attack (AOA) towards the tip which leads to a change in magnitude and rotation of the lift and drag force. The induced drag due to the lift vector rotation causes a larger
reduction of the tangential forces than the normal forces towards the tip which agrees well with the observed anisotropy of the tip loss factor by Wimshurst and Willden (2017).

Due to the smearing of the blade forces and the typically coarse grid resolution near the blades, a tip loss correction is also needed for actuator line computations. The modification described in this article can be used for these simulations in the same way as for an AD setup.

In the following section, the main equations of Shen's original tip loss correction are briefly introduced. In Sect. 3, an alternative computation of the tangential force is derived. Finally, the new tip loss correction is applied in AD computations, and compared to BEM computations in Sect. 5.

## 2 Shen's original tip correction

A common tip correction for a CFD AD model based on airfoil data is the tip correction of Shen et al. (2005), which multiplies
the tangential $c_t$ and normal $c_n$ force coefficients with a factor $F_1^{\mathrm{Shen}}$:

$$c_t^r = F_1^{\mathrm{Shen}} c_t, \qquad c_n^r = F_1^{\mathrm{Shen}} c_n \tag{1}$$

$F_1^{\mathrm{Shen}}$ is based on the tip loss function of Glauert (1963).

$$F_1^{\mathrm{Shen}} = \frac{2}{\pi} \cos^{-1} \left[ \exp \left( -g \frac{B(R-r)}{2r \sin \phi} \right) \right], \qquad g = \exp \left[ -c_1 (B\lambda - c_2) \right] + 0.1 \tag{2}$$

where $B$ is the number of blades, $R$ is the blade radius, $r$ is the radial coordinate, $\phi$ is the flow angle, $\lambda$ is the tip speed ratio,
and $c_1$ and $c_2$ are constants, where $c_1 = 0.125$ and $c_2$ is often used to fit the blade forces from RANS with a reference.

## 3 A modified tip correction

A modification to the tip loss correction of Shen et al. (2005) is described in this section. The modification relies on a series of assumptions.





### 3.1 Assumptions

1. The axial force can be matched well using Shen's tip loss correction.

2. The inflow angle is small.

3. Following from the small inflow angle: the influence of the tip vortex on the relative velocity close to the tip is small. Thus the change in lift is dominated by the change in angle of attack, not the change in relative velocity.

4. The flow close to the tip is attached with a constant lift gradient.

5. The drag contribution on the thrust force is small compared to the lift contribution: $D \sin \varphi << L \cos \varphi$ with the inflow angle $\varphi$.

All of these assumptions are true in normal operation up to rated wind speed. At high wind speed, the inflow angle towards the tip will become large and the outer airfoil sections might approach negative stall. In these conditions, the load distribution is changed, with the highest load inboard and strong vorticity trailed from the mid part of the blade, and a conventional tip loss correction is not meaningful. In high wind conditions, modeling trailed vorticity is necessary to bring the load distribution closer to CFD (Pirrung et al., 2016). Therefore, the assumptions hold in the operating conditions where a tip loss correction is applicable.

### 3.2 Derivation

The factor $F_1^{\text{Shen}}$ that scales the normal force, see Eq. (1), can be translated into a corresponding change in AOA $\Delta\alpha$, see Fig. 1. Assuming small inflow angles, this change in AOA will lead to an equivalent normal load distribution as applying the factor $F_1$. The induced drag due to the rotation of the lift force leads to a smaller resulting tangential force which is in agreement with the stronger tangential tip loss observed by Wimshurst and Willden (2018).

Assuming a small inflow angle and neglecting the drag influence, which is the product of two small terms, the normal force $F_{N,2D}$ can be approximated by the lift force $L_{2D}$:

$$F_{N,2D} = L_{2D} \cos \varphi + D_{2D} \sin \varphi \approx L_{2D} = \frac{\rho c}{2} v_{rel}^2 \frac{\partial C_L}{\partial \alpha}(\alpha - \alpha_0) \tag{3}$$

The difference in normal force is assumed to be caused by a change in AOA due to the trailed vorticity at the blade tip:

$$F_{N,3D} \approx F_1 L_{2D} \approx \frac{\rho c}{2} v_{rel}^2 \frac{\partial C_L}{\partial \alpha}(\alpha - \Delta\alpha - \alpha_0) \tag{4}$$

Inserting Eq. (3) in Eq. (4) leads to an expression for the change in AOA $\Delta\alpha$:

$$\Delta\alpha = (\alpha - \alpha_0)(1 - F_1) \tag{5}$$

The change in inflow angle is identical with $\Delta\alpha$. Thus the tangential force at the changed inflow angle be computed as:

$$F_{T,3D} = L F_1 \sin(\varphi - \Delta\alpha) - D(\alpha - \Delta\alpha) \cos(\varphi - \Delta\alpha) \tag{6}$$



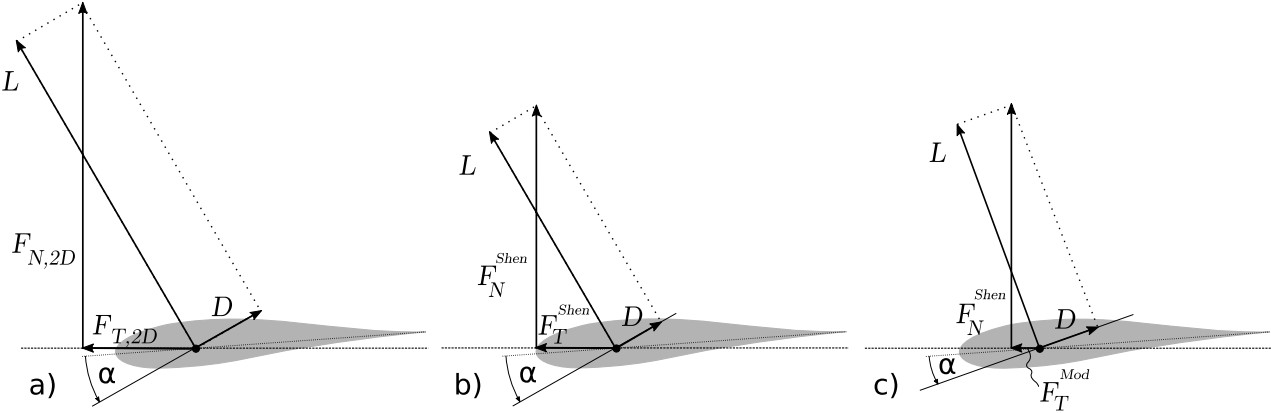

**Figure 1.** Sketches of the aerodynamic forces at an airfoil. The angles are exaggerated and the lift to drag ratio is displayed unrealistically low for clarity. Sketch a) illustrates the aerodynamic forces before applying a tip loss correction. When applying a tip loss correction by Shen et al. (2005), all forces are scaled with a certain factor as shown in b). Instead, a change in AOA $\alpha$ can be defined that produces the same normal force $F_N^{Shen}$, but leads to a much reduced tangential force $F_T^{Mod}$, c).

## 4 Methodology

### 4.1 Test cases

The modification of the tip loss correction will be applied in five different test cases, see Tab. 1. Cases 1-3 represent operation at the design tip speed ratio for different rotors. The Vestas V29 rotor has recently been installed in a 4-rotor configuration known as the Babylon multi-rotor wind turbine, the NREL 5MW is a reference wind turbine developed by the National Renewable Energy Laboratory Jonkman et al. (2009) and the MEXICO turbine has been used in wind tunnel tests and a range of code comparisons Boorsma and Schepers (2018). The additional Cases 4 and 5 are computations of the V29 rotor at higher wind speeds. At these higher wind speeds, the load distribution differs from the load distribution at design tip speed ratio and the tip loss corrections in BEM are less valid. These cases are included to show that the modification described in this paper doesn't introduce erratic behaviour when applied outside of design conditions.

**Table 1.** Test cases.

| Case | Rotor | $D$ [m] | $U$ [m/s] | TSR | Pitch [°] |
|---|---|---|---|---|---|
| 1 | V29 | 29.2 | 7 | 7.6 | -0.6 |
| 2 | NREL-5MW | 126.0 | 8 | 7.506 | 0.0 |
| 3 | MEXICO | 4.5 | 15.06 | 6.7 | -2.3 |
| 4 | V29 | 29.2 | 12 | 5.266 | 3.37 |
| 5 | V29 | 29.2 | 18 | 3.508 | 16.72 |



## 4.2 HAWC2

The BEM model implemented in the HAWC2 aeroelastic code (Larsen and Hansen, 2007) is used as a reference in this article. A tip loss correction by Wilson and Lissaman (1974) is implemented in the code. Because all the cases investigated in this article are in uniform inflow, the additional aerodynamic sub models to handle for example shear, yaw and dynamic inflow are

not relevant. The aerodynamic sections, in the present study 50 (V29) and 60 (NREL 5MW, Mexico rotor), are placed using a cosine distribution. This ensures a good resolution at the tip. All computations are performed on stiff rotors.

Using a BEM model as a reference has the advantage that it is essentially a disc model that relies on exactly the same airfoil data as the AD model. This makes comparing the results more clear than if a more complex aerodynamic reference was chosen. The BEM model in HAWC2 has been compared to high-fidelity codes many times. A comparison for the NREL 5MW

in uniform inflow by Madsen et al. (2012) showed very good agreement of the loading obtained by HAWC2 with results from CFD and vortex wake codes. A recent comparison with measurements and codes of all fidelity levels for the Mexico rotor is described by Boorsma et al. (2018).

## 4.3 EllipSys3D RANS

EllipSys3D is an incompressible finite volume flow solver initially developed by Sørensen (1994) and Michelsen (1992). Since

the flow variables are located at the cell centers a Rhie and Chow (1983) algorithm is applied, which is used in a modified form to avoid pressure-velocity decoupling when additional momentum body forces are employed (Réthoré and Sørensen, 2012). The equations are solved by a SIMPLE algorithm (Patankar and Spalding, 1972) and the convective terms are discretized by a QUICK scheme (Leonard, 1979). In this article, we use a Reynolds-averaged Navier-Stokes (RANS) method where the turbulence is modelled by the $k$-$\omega$ SST turbulence model of Menter (1993). The wind turbine rotor is represented as an AD

where the forces are computed from airfoil data as used by Réthoré et al. (2014). The AD grid is a polar grid with 95 and 180 grid points in the radial and azimuthal directions, respectively. The AD is located at $\{x,y,z\} = \{2D, 1.5D, 1.5D\}$ in a Cartesian flow domain as depicted in Fig. 2, where $D$ is the rotor diameter. In the center of the flow domain, a uniform spaced domain of dimensions $14 \times 3 \times 3D^3$ is applied with a grid spacing of $D/40$. A grid study refinement study is presented in Sect. 5.1 that motivates a $D/40$ grid spacing. Around the uniformly spaced domain, the grid is stretched outwards over

a distance of $100D$ in all directions, which results in a total flow domain size of $214 \times 203 \times 203D^3$ in the streamwise, and two cross directions, respectively. A uniform flow at the inflow boundary at $x = -100D$ is specified. The four lateral boundaries ($y = z = -100D$ and $y = z = 103D$ are periodic and a Neumann condition for the all the gradients in the direction perpendicular to the outflow boundary at $x = 114D$ are used. The molecular viscosity is set such that a Reynolds (based on the rotor diameter) of 1 million is obtained. To avoid an influence of the inlet eddy-viscosity on the flow solution, we make sure

that the eddy viscosity is much smaller than the molecular viscosity by setting the turbulent kinetic energy and dissipation as $10^{-2}$ m$^2$/s$^2$ and $10^6$ 1/s at the inlet boundary, respectively.



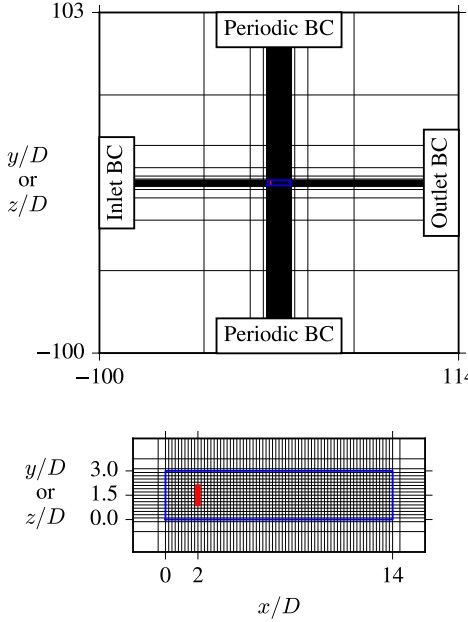

**Figure 2.** Numerical domain of RANS simulations where one in every eight cells are shown. Bottom plot is a zoomed view around the AD. Blue lines indicate the uniform spaced domain with a cell size of $D/40$. Red line represent the AD.

## 4.4 Procedure for testing the modified tip correction

The modified tip correction for the tangential load distribution proposed in Sect. 3 only works well if the axial load distribution is correct. However, we find that the shape and the number of constants of the original tip correction function of Shen et al. (2005) (Eq. 2) is not sufficient to match the axial load distribution of HAWC2 for Cases 2 and 3. In order to shown that the

5   modified tip correction for the tangential load distribution works well, we propose an additional tip correction function that has an additional constant $h$ and a different shape:

$$F_1^{\text{Test}} = \frac{2}{\pi} \cos^{-1} \left[ (1-h) \exp \left( -g \frac{B(R-r)}{2r\sin\phi} \right) + h \right] + \left( 1 - \frac{2}{\pi} \cos^{-1}[h] \right) \tag{7}$$

The constant $h$ should be chosen between 0 and 1. For $h = 0$, the original tip correction function of Shen et al. (2005) is obtained. Following Wimshurst and Willden (2018), we fit the two tip correction functions (Eqns. 2 and 7) with a calculated

10   reference tip correction function for both the normal $F_{1,n}^{\text{Ref}}$ and tangential $F_{1,t}^{\text{Ref}}$ load distributions:

$$F_{1,n}^{\text{Ref}} = \frac{F_n^{\text{HAWC2}}}{F_n^{\text{RANS}}}, \qquad F_{1,t}^{\text{Ref}} = \frac{F_t^{\text{HAWC2}}}{F_t^{\text{RANS}}} \tag{8}$$

where the load distributions from RANS are computed without a tip correction. The fit is carried out for the last 30% of the rotor radius using a least-squares approach and the results are depicted in Fig. 3 for Cases 1-3. A perfect fitted tip correction does not guarantee a simulated force distribution in RANS that matches the reference because of the interaction of the forces



with the local flow. One could use an iterative procedure to match the reference force distributions exactly; however, in practice the simulated force distributions in RANS (using the fit of Fig. 3) produces a reasonable match with the reference force distributions.

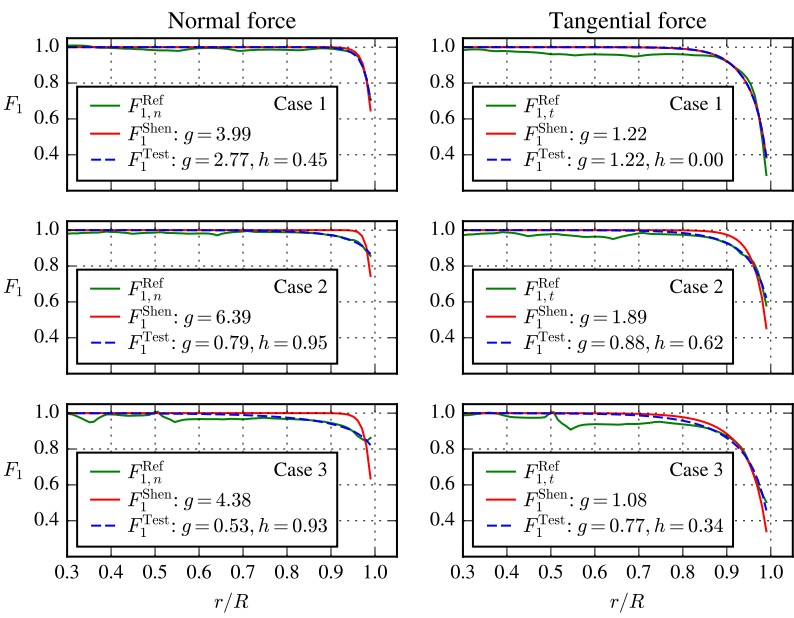

**Figure 3.** Fit of tip correction functions.

Fig. 3 shows that both tip correction functions are very similar for Case 1, especially for the tangential load distribution.
5  However, Cases 2 and 3 show that the shape of the original tip correction of Shen et al. (2005) cannot be fitted with the reference for the axial force distributions and has minor errors for the tangential force. On the contrary, the test tip correction function of Eq. 7 can be fitted with the reference using the additional constant $h$ for both the axial and tangential force distributions for all three cases. Fig. 3 shows that the tip correction function for tangential force distribution should be different from the tangential force distribution, as also shown by Wimshurst and Willden (2018). This is the main motivation for our proposed modification
10  for the tangential force when using the tip correction of Shen et al. (2005) from Sect. 3. The test tip correction function of Eq. (7) that is fitted with the reference normal force distribution will be used to show that the proposed modification for the tangential force works if the normal force distribution matches. It also possible to simply use the tip correction function of Eq. (7) for both the axial and the tangential load distributions where different values of $g$ and $h$ are used for each force distribution.

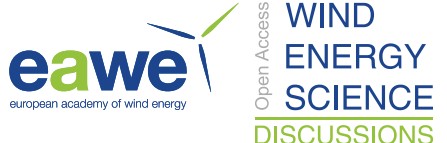

## 5 Results

### 5.1 Grid refinement study of RANS simulations

The influence of the grid spacing around the AD on the normal and tangential blade force distributions is shown in Fig. 4 for Case 1. The modified tip correction as proposed in Sect. 3 is employed using $C_2 = 29$. Fig. 4 shows that both the normal and
5   tangential blade force distributions converge with grid refinement and the difference between a grid spacing of $D/40$ and $D/80$ is negligible. Tab. 2 shows the influence of the grid spacing on the integrated forces in the form of the power coefficient $C_P$ and the thrust coefficient $C_T$. In addition, an extrapolated value is shown calculated by a mixed order analysis as introduced by Roy (2003) and also applied in Réthoré et al. (2014). The extrapolated value is used to estimate the discretization error. Tab. 2 shows that a grid spacing of $D/20$ is fine enough to calculate the power and thrust coefficients within an error of 0.3%,
10   and could be considered fine enough. However, in the present work, we choose to use a grid spacing of $D/40$, such the force distributions at the blade tip (the focus in this article) are grid independent, as shown in Fig. 4.

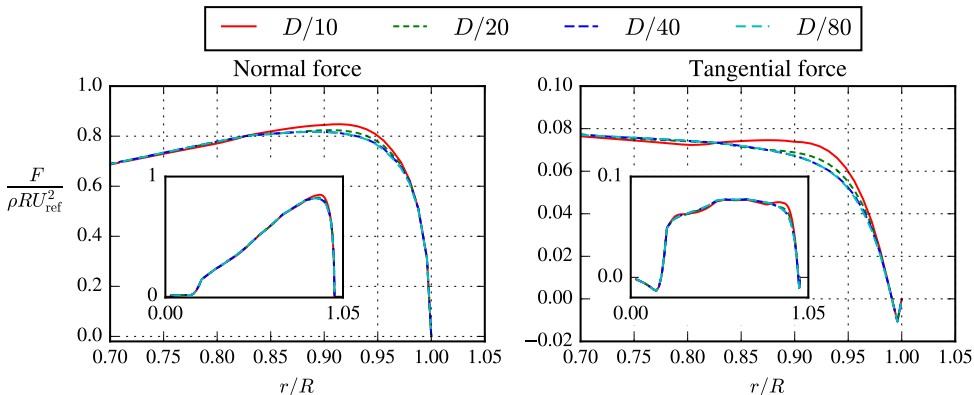

**Figure 4.** Grid refinement study of RANS simulations for Case 1 using modified tip correction. Inner plots are zoomed out views.

**Table 2.** Grid refinement study of RANS simulations for Case 1 using modified tip correction.

|              | $D/10$ | $D/20$ | $D/40$ | $D/80$ | Extrapolated |
|--------------|--------|--------|--------|--------|--------------|
| $C_P$ value  | 0.4699 | 0.4616 | 0.4606 | 0.4604 | 0.4603       |
| $C_P$ error  | 2.1%   | 0.28%  | 0.06%  | 0.02%  |              |
| $C_T$ value  | 0.8151 | 0.8103 | 0.8098 | 0.8097 | 0.8095       |
| $C_T$ error  | 0.69%  | 0.10%  | 0.04%  | 0.02%  |              |



## 5.2 Comparison of tip corrections.

A comparison of the load distributions for Cases 1-3, see Tab. 1, is shown in Fig. 5. The reference result is obtained from a HAWC2 BEM computation. The other lines are results of AD simulations using the setup as described in Sect. 4.3. The forces are normalized with the air density $\rho$ multiplied by the rotor radius $R$ and the squared free stream velocity $U_{ref}^2$ The different tip corrections shown in the plot are:

$F_1^{\text{Shen}}$: $F_n$ fit: The tip loss for $F_n$ and $F_t$ is computed using Eq. (2). The parameters are chosen to fit $F_n$.

$F_1^{\text{Test}}$: $F_n$ fit: The tip loss for $F_n$ and $F_t$ is computed using Eq. (7). The parameters are chosen to fit $F_n$.

$F_1^{\text{Shen}}$: $F_n$ fit $+ F_t$ mod: The tip loss for $F_n$ is computed using Eq. (2). $F_t$ is computed using Eq. (6).

$F_1^{\text{Test}}$: $F_n$ fit $+ F_t$ mod: The tip loss for $F_n$ is computed using Eq. (7). $F_t$ is computed using Eq. (6).

$F_1^{\text{Test}}$: $F_n + F_t$ fit: The tip loss for $F_n$ and $F_t$ is computed independently using Eq. (7).

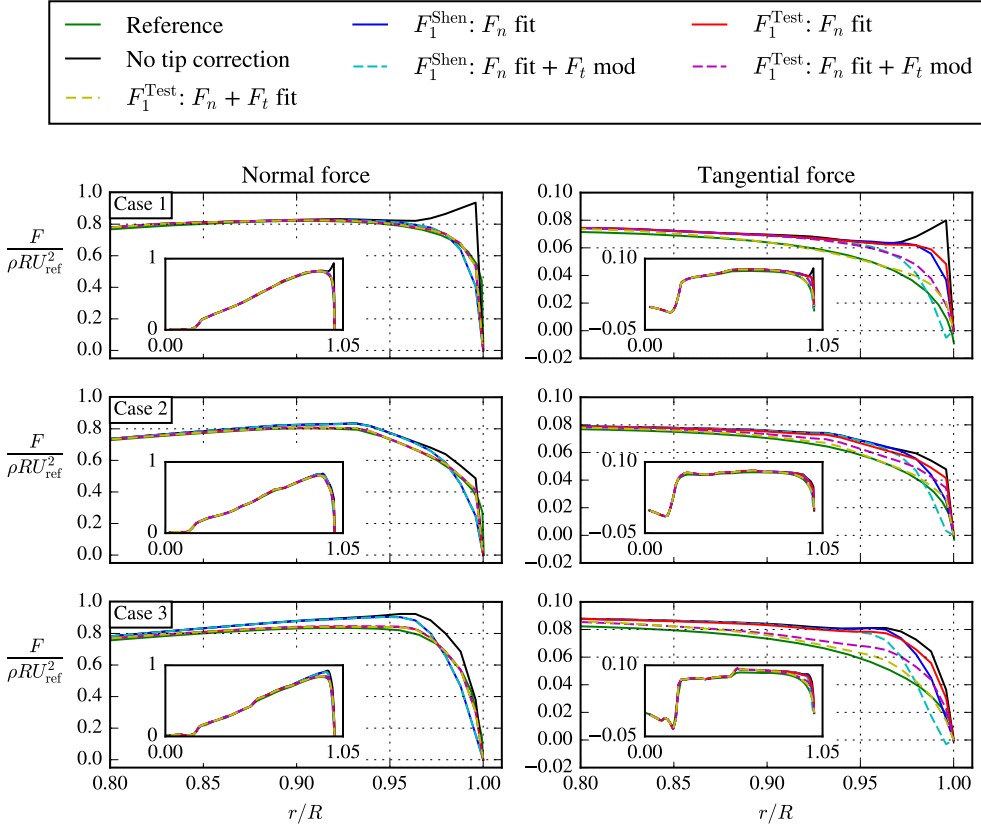

**Figure 5.** Comparison of tip corrections for Cases 1-3. Inner plots are zoomed out views.





The tip loss factor by Shen et al. (2005) fits the normal force quite well for the V29 rotor (Case 1), see the top left plot of Fig. 5. There is some under prediction at the outmost percent of the span where the fit is improved with the $F_1^{\text{Test}}$ tip loss factor. Applying the same tip loss correction factor on both normal and tangential force leads to a significant over prediction of the tangential force. The modified tangential force, Eq. (6), agrees better with the BEM reference. However, the

5     under prediction of $F_n$ close to the tip with Shen's tip loss factor is also seen in in the tangential force. Generally the tangential force in the tip region is over predicted in the AD simulations and can only get close to the BEM results by using a dedicated fit of Eq. 7.

Similar conclusions hold for the MEXICO and NREL 5MW rotors (Cases 2 and 3, respectively). The tip correction factor by Shen et al. (2005) fits less well for these rotors and the under predicted normal force towards the tip is amplified when the

10    modified tangential force is used in both rotors. When a good fit of the normal force is ensured using Eq. (7) the modification in Eq. (6) reduces the error on the tangential force significantly. A dedicated, independent tip loss factor for the tangential force leads to the best results.

The load distributions for the V29 rotor at higher wind speeds, Cases 4 and 5 in Tab. 1, are shown in Fig. 6. These cases are included to investigate the behavior of the modified tip loss correction when the underlying assumptions are violated due to

15    large inflow angles and lower lift to drag ratios at the tip at high wind speeds. It can be seen that there is still an improvement at 12 m/s (top plot) and the modification doesn't significantly change the tangential loads at 18 m/s. This is because the tip loss effect is much weaker at high wind speeds, where the tip is deloaded and most of the power and thrust are generated on the inboard blade sections.

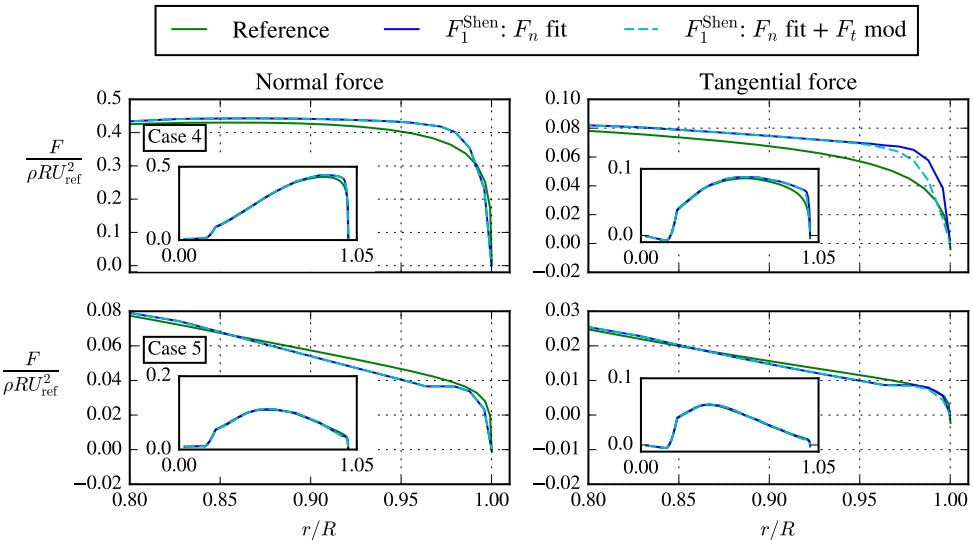

**Figure 6.** Comparison of tip corrections for Cases 4 and 5. Inner plots are zoomed out views.



## 6 Conclusions

A modification of the tangential loading when applying a commonly used tip loss factor for AD simulations was proposed in this article. This modification addresses the differences in the tip loss effect for normal and tangential forces that have previously been observed in both low-fidelity and high fidelity computations. The modeled mechanism is the rotation of the
5  lift force due to the velocity that is induced by the tip vortex.

The modification has been applied on three different rotors and leads to improved tangential loads when compared to a reference BEM. This improvement becomes more clear when the fit for the axial loading is good. To ensure this, an alternative equation for the tip loss factor with one additional parameter is proposed. Using this equation independently for both normal and tangential forces leads to the best agreement between AD and reference BEM simulations. No problematic behavior at
10  higher wind speed was observed when applying the modified tip loss correction for the tangential force.

The proposed modification is easy to implement and comes at a negligible computational cost. A refitting of parameters is not required if a good fit of the normal force could be obtained using the original tip loss model.

*Acknowledgements.* This is work is sponsored by Vestas Wind System A/S





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
