# Peer review of "A simple improvement of a tip loss model for actuator disc and actuator line simulations"

_Wind Energy Science, 2018_

## Referee Comment (RC1) · Anonymous Referee #1 · 30 Oct 2018

A simple improvement of a tip loss model for actuator disc and actuator line simulations

This paper presents a possible improvement for Shen's tip loss correction which takes into account tangential induction. The idea is interesting, however the explanations, derivations and results in the manuscript are unclear. The manuscript is also missing many important details. The following suggestions should be addressed before the manuscript can be considered for publication in WES.

It is difficult to assess the validity of the correction presented when it is only compared to another tip loss correction (Wilson and Lissaman).

There is no physical insight for why the new correction has a term "h".

[Figure]

A description of the actuator disk model is missing.

Actuator line model is mentioned but it is never used? I doubt that this correction would work on the actuator line model because it does not take into account the part of the tip vortex that is resolved. It seems the model will only work if there is no resolved tip vortex at all.

Derivation: The derivation presented is not clear.

I do not see how equation 5 comes from equations 3 and 4.

This sentence is unclear: "The change in inflow angle is identical with dalpha"

Table 1: Use the same number of significant digits.

Section 5.2 – This section is unclear. What is the difference between all these corrections, and which one should be used? All the results are quite different. What is the reference? Are results supposed to match the reference?

I'm sure that there is a physical insight to the model presented? Please describe this.

---

## Referee Comment (RC2) · Anonymous Referee #2 · 9 Nov 2018

Review of paper WES-2018-59

A simple improvement to a tip loss model for actuator disc and actuator line simulations

Authors: G. R. Pirrung and Maarten P. van der Laan

The paper presents a simple-to-implement correction to the tangential tip loss model/factor of Shen et al. (2005) for use in actuator disc (AD) and actuator line (AL) simulations. The new model is described and implemented, and quantitative comparisons of tip loss factor and spanwise distributions of normal-/tangential forces are performed against a reference blade-element momentum (aeroelastic) code, HAWC2. The work is original and potentially important to the wind energy science community.

[Figure]

General comments: In its present form, the paper is not written in a clear and concise style. There are a few typos in the manuscript that are relatively minor; however, the presentation and physical reasoning for the proposed tangential correction are both weak and unclear. No comparisons to actual experimental (or blade-resolved CFD) data are provided, though these are available from other publications by authors' colleagues at the same institution (?) and instrumental in supporting evidence that the proposed model captures the associated physics. It is therefore that the reviewer is not convinced that the proposed tangential correction is of any use to the wind energy community. Also, the tangential correction is argued to be of use not only in AD but also AL computations, with the latter one not included in the comparisons. The authors further base their model on recent work by Wimshurst and Willden (2017,2018) and do not provide an adequate literature review and benchmarking against other approaches that have been proposed in recent years.

RECOMMENDATION: REJECT - The paper is not acceptable in its present form, style, discussion of (incomplete) results, and lack of quantitative comparisons against measured data and CFD results available in the literature. This is unacceptable in a reputed scientific journal.

To make the manuscript acceptable requires changes that go beyond a typical major revision in a reputed wind energy science journal. The reviewer encourages the authors to perform considerable more work on quantitative comparisons against experimental data and available CFD simulations, see detailed comments below, and resubmit as a new paper.

1. MAJOR CONCERN There is no experimental data given for comparison. This is unacceptable, particularly if the authors want to make a case that their proposed tangential correction is of use to the wind energy community. In addition, only Case 3 refers to an actual (MEXICO) experiment. The reviewer encourages the authors to include the standard MEXICO cases at 10,15,24 m/s and also the NREL Phase VI rotor. Then comparisons can be made against e.g. the NREL 5-MW or DTU-10MW turbines
where a number of CFD simulations are available in the literature for comparison.

2. MAJOR CONCERN The title claims that the proposed tangential correction factor to be applied to the model of Shen et al. is also useful for AL simulations. This has to be shown with quantitative comparisons. It seems that the authors could conduct those using the EllipSys3D code.

3. MAJOR CONCERN The description of the new tangential correction factor is weak and does not seem to be rooted in any physics. What does the 'h' term describe ? What physical phenomenon is captured ?

Other Comments:

- Abstract: The reviewer does not necessarily agree that a separate tangential correction is necessary. A model rooted in the driving flow physics should be as easy and elegant as possible.

- Introduction (page 1): Not all AL models require a tip correction. This does not mean that using a tip correction is incorrect, but it should be acknowledged that there are other approaches.

- Introduction (page 2): The review of recent literature is incomplete. Some work has been published on the de-cambering effect and using free-wake method results as a look-up table for improved tip corrections. This should be acknowledged and it should be clarified what similar or other physics the proposed model captures.

- Introduction (page 2, last sentence): Why only AD computations ? Inconsistent with manuscript title. Also, the comparisons are meaningless in the absence of experimental data and blade-resolved CFD simulations.

- Pages 2 and 3: Not sue if it is necessary to repeat descriptions of the models by Shen et al. and Wimshurst and Willden. It would make sense having the proposed model being described in conjunction with the older models (and not later on page 6)

[Figure]

- Table 1: Inadequate choice of test cases, see major comments above.

- Page 6, section 4.4: What is the physics behind 'h' ?

- Figure 3: There need to be available measured or CFD data for comparisons. How do differences between tip models integrate to deltas in thrust and power ?

- Page 8: Do not start sentence with 'Tab.' or 'Fig.'. More examples throughout the manuscript.

- Page 9, section 5.2: Justify why these model variations have been chosen. What is the effect on integrated thrust and power ?

- Figure 5: What is 'Reference' ? Add HAWC2 BEM to legend so this is not confusing. This figure (as are others) is of no use as there are no data for comparison. In particular, one would hope for improved comparisons against data for the proposed new model and tangential forces.

- Page 11, Conclusions: The second paragraph is unclear and not precise, e.g. "... when the fit for the axial loading is good".

- Page 11, Conclusions: The last sentence of the manuscript is very concerning as it states that "... refitting is not required if a good fit of the normal force could be obtained using the original tip loss model" How can one know what to choose a-priori, if the method is used as a predictive tool ?

---

## Author Comment (AC1) · 11 Dec 2018

**Reviewer 1**

*Thank you for the review. We apologize if the article was unclear. There appears to be a misunderstanding about the proposed tip loss correction. We propose to compute the tangential loading according to Equation (6) to take into account the rotation of the resulting force due to the increasing induction towards the tip. This can be understood as an induced drag effect. Treating the tangential loading in this way adds some physics to the modeling of the tip loss effect without needing any additional parameters. Instead the effect of tip loss on the tangential loading follows directly from the axial loading. This requires a good match of the axial loading.*

*To obtain an improved match of the axial loading we introduced the expression in Equation (7) of the article. This Equation is not based on physics but purely on curve fitting. We do not propose to employ this Equation as a tip loss correction. Instead we only use it to show that the proposed method in Equation (6) will correctly modify the tangential loading if the axial forces agree to a reference.*

*We will adapt the article to better include the above explanations. We will also remove the 'actuator line' reference from the title of the article. Applying a tip loss method on actuator line methods is a controversial topic that will need some more detailed studies in the future.*

A simple improvement of a tip loss model for actuator disc and actuator line simulations This paper presents a possible improvement for Shen's tip loss correction which takes into account tangential induction.

- *This is a misunderstanding. We take into account the effect of the axial induction on the tangential forces. Tangential induction is not mentioned in the article.*

The idea is interesting, however the explanations, derivations and results in the manuscript are unclear. The manuscript is also missing many important details. The following suggestions should be addressed before the manuscript can be considered for publication in WES.

It is difficult to assess the validity of the correction presented when it is only compared to another tip loss correction (Wilson and Lissaman). There is no physical insight for why the new correction has a term "h".

- *There seems to be a misunderstanding. The tip loss correction we present consists of Equations (5) and (6), not of Equation (7). Equation (7), that includes the term 'h' is only used to obtain a good fit of axial and tangential forces for comparison purposes. We apologize if this was not made clear in the original submission. We will clarify this in the revised version of the article.*

A description of the actuator disk model is missing.

- *The description of the CFD method is in Section 4.3. The listed references Réthoré and Sørensen (2012), and Réthoré et al. (2014) describe the actuator disk method in more detail.*

Actuator line model is mentioned but it is never used? I doubt that this correction would work on the actuator line model because it does not take into account the part of the tip vortex that is resolved. It seems the model will only work if there is no resolved tip vortex at all.

- *Actuator line models do need tip corrections because the tip vortex is not fully resolved. Only for a weak tip vortex and a very fine grid resolution one could choose to not use a tip correction. Shen's tip correction is used for actuator line models in:*
  - *Wimshurst, A. and Willden, R. H. J.: Analysis of a tip correction factor for horizontal axis turbines, Wind Energy, 20, 1515-1528, DOI: 10.1002/we.2106*
  - *Breton, S. P., Shen, W. Z., and Ivanell, S. (2017). Validation of the actuator disc and actuator line techniques for yawed rotor flows using the New Mexico experimental data. Journal of Physics: Conference Series, 854, [012005]. DOI: 10.1088/1742-6596/854/1/012005*

Derivation: The derivation presented is not clear. I do not see how equation 5 comes from equations 3 and 4.

- *Here is the full derivation: The last part of Equation (3) in the article is:*

$$L_{2D} = \frac{\rho c}{2} v_{rel}^2 \frac{\partial C_L}{\partial \alpha} (\alpha - \alpha_0) \tag{0.1}$$

*Equation (4) in the article:*

$$F_{N,3D} \approx F_1 L_{2D} \approx \frac{\rho c}{2} v_{rel}^2 \frac{\partial C_L}{\partial \alpha} (\alpha - \Delta\alpha - \alpha_0) \tag{0.2}$$

*Inserting Eq. (0.1) in Eq. (0.2):*

$$F_1 \frac{\rho c}{2} v_{rel}^2 \frac{\partial C_L}{\partial \alpha} (\alpha - \alpha_0) \approx \frac{\rho c}{2} v_{rel}^2 \frac{\partial C_L}{\partial \alpha} (\alpha - \Delta\alpha - \alpha_0) \tag{0.3}$$

$$F_1 (\alpha - \alpha_0) \approx (\alpha - \Delta\alpha - \alpha_0) \tag{0.4}$$

$$\Delta\alpha \approx (\alpha - \alpha_0)(1 - F_1) \tag{0.5}$$

This sentence is unclear: "The change in inflow angle is identical with dalpha"

- *It means $\Delta\varphi = \Delta\alpha$. We will add that equation in parentheses after the sentence.*

Table 1: Use the same number of significant digits.

- *We have now used two significant digits for all percentages.*

Section 5.2 – This section is unclear. What is the difference between all these corrections, and which one should be used? All the results are quite different. What is the reference? Are results supposed to match the reference?

- *We will change the legend in the figures so that they state 'Reference: HAWC2 BEM'. We will remove the last line in Figure 5 ($F_1^{Test}$: $F_n + F_t$ fit:) where we fit the axial and tangential forces independently, because it is confusing and not necessary for the conclusions of the article. We have extended the descriptions of the remaining methods:*

  $F_1^{Shen}$: $F_n$ fit: The tip loss for $F_n$ and $F_t$ is computed using Eq. (2). The parameters are chosen to fit $F_n$. This corresponds to using Shen's tip loss correction

  $F_1^{Test}$: $F_n$ fit: The tip loss for $F_n$ and $F_t$ is computed using Eq. (7). The parameters are chosen to fit $F_n$. The purpose of this to obtain a better agreement of the axial force. This is possible because the 'test' function has an additional parameter.

  $F_1^{Shen}$: $F_n$ fit + $F_t$ mod: The tip loss for $F_n$ is computed using Eq. (2). $F_t$ is computed using Eq. (6). This means that the modified tip loss correction proposed here is used together with Shen's tip loss correction.

  $F_1^{Test}$: $F_n$ fit + $F_t$ mod: The tip loss for $F_n$ is computed using Eq. (7). $F_t$ is computed using Eq. (6). Thus the modified tip loss correction is based on a more closely matching axial force. Based on this the quality of the tangential load correction can be investigated with less error progression from the axial load correction.

I'm sure that there is a physical insight to the model presented? Please describe this.

- *The most concise description of the physical basis for the model (Equations (5) and (6)) is found in the conclusions: 'The modeled mechanism is the rotation of the lift force due to the velocity that is induced by the tip vortex.'. The derivation and the more detailed physical reasoning is found in Section 3 of the article.*

---

## Author Comment (AC2) · 11 Dec 2018

**Reviewer 2**

*Thank you for the review. We apologize if the article was unclear. There appears to be a misunderstanding about the proposed tip loss correction. We propose to compute the tangential loading according to Equation (6) to take into account the rotation of the resulting force due to the increasing induction towards the tip. This can be understood as an induced drag effect. Treating the tangential loading in this way adds some physics to the modeling of the tip loss effect without needing any additional parameters. Instead the effect of tip loss on the tangential loading follows directly from the axial loading. This requires a good match of the axial loading.*

*To obtain an improved match of the axial loading we introduced the expression in Equation (7) of the article. This Equation is not based on physics but purely on curve fitting. We do not propose to employ this Equation as a tip loss correction. Instead we only use it to show that the proposed method in Equation (6) will correctly modify the tangential loading if the axial forces agree to a reference.*

*We will adapt the article to better include the above explanations. We will also remove the 'actuator line' reference from the title of the article. Applying a tip loss method on actuator line methods is a controversial topic that will need some more detailed studies in the future.*

The paper presents a simple-to-implement correction to the tangential tip loss model/factor of Shen et al. (2005) for use in actuator disc (AD) and actuator line (AL) simulations. The new model is described and implemented, and quantitative comparisons of tip loss factor and spanwise distributions of normal-/tangential forces are performed against a reference blade-element momentum (aeroelastic) code, HAWC2. The work is original and potentially important to the wind energy science community.

General comments: In its present form, the paper is not written in a clear and concise style. There are a few typos in the manuscript that are relatively minor; however, the presentation and physical reasoning for the proposed tangential correction are both weak and unclear. No comparisons to actual experimental (or blade-resolved CFD) data are provided, though these are available from other publications by authors' colleagues at the same institution (?) and instrumental in supporting evidence that the proposed model captures the associated physics. It is therefore that the reviewer is not convinced that the proposed tangential correction is of any use to the wind energy community. Also, the tangential correction is argued to be of use not only in AD but also AL computations, with the latter one not included in the comparisons. The authors further base their model on recent work by Wimshurst and Willden (2017,2018) and do not provide an adequate literature review and benchmarking against other approaches that have been proposed in recent years. RECOMMENDATION: REJECT - The paper is not acceptable in its present form, style, discussion of (incomplete) results, and lack of quantitative comparisons against measured data and CFD results available in the literature. This is unacceptable in a reputed scientific journal.

To make the manuscript acceptable requires changes that go beyond a typical major revision in a reputed wind energy science journal. The reviewer encourages the authors

to perform considerable more work on quantitative comparisons against experimental data and available CFD simulations, see detailed comments below, and resubmit as a new paper.

1. MAJOR CONCERN There is no experimental data given for comparison. This is unacceptable, particularly if the authors want to make a case that their proposed tangential correction is of use to the wind energy community. In addition, only Case 3 refers to an actual (MEXICO) experiment. The reviewer encourages the authors to include the standard MEXICO cases at 10,15,24 m/s and also the NREL Phase VI rotor. Then comparisons can be made against e.g. the NREL 5-MW or DTU-10MW turbines where a number of CFD simulations are available in the literature for comparison.

- *It is correct that the article contains only comparisons against BEM code results. But for the following reasons we do not think that comparisons with experiment or CFD are necessary in the present article:*

  - *The comparison actuator disc vs BEM is a quite fair comparison because both are disc models that do not inherently model the induced velocity at the individual blades.*

  - *Both the actuator disc method and BEM use exactly the same airfoil data. A comparison to CFD or experiment, which do not rely on airfoil data, will introduce large uncertainties.*

  - *The HAWC2 BEM code has already been compared to CFD, measurements and vortex codes in the literature. We do not see that such a comparison would add much value to the present article.*

2. MAJOR CONCERN The title claims that the proposed tangential correction factor to be applied to the model of Shen et al. is also useful for AL simulations. This has to be shown with quantitative comparisons. It seems that the authors could conduct those using the EllipSys3D code.

- *This is correct. We are aware that scientists use the model of Shen et al. also for AL simulations:*

  - *Wimshurst, A. and Willden, R. H. J.: Analysis of a tip correction factor for horizontal axis turbines, Wind Energy, 20, 1515-1528, DOI: 10.1002/we.2106*

  - *Breton, S. P., Shen, W. Z., and Ivanell, S. (2017). Validation of the actuator disc and actuator line techniques for yawed rotor flows using the New Mexico experimental data. Journal of Physics: Conference Series, 854, [012005]. DOI: 10.1088/1742-6596/854/1/012005*

  *In AL cases the rotation of the resulting force should yield the same benefit as for the AD cases. It is true that - depending on the discretization - some of the tip vortex is resolved in AL computations and thus the applicability of a tip loss correction is not totally clear. We will remove the references to actuator line from the article (including the title).*

3. MAJOR CONCERN The description of the new tangential correction factor is weak and does not seem to be rooted in any physics. What does the 'h' term describe ? What physical phenomenon is captured ?

- *It seems that there is a misunderstanding. The tip loss correction we present consists of Equations (5) and (6), not of Equation (7). Equation (7) is only used to obtain a good fit of axial and tangential forces for comparison purposes.*

- *The most concise description of the physical basis for the model (Equations (5) and (6)) is found in the conclusions: 'The modeled mechanism is the rotation of the lift force due to the velocity that is induced by the tip vortex.'. The derivation and the more detailed physical reasoning is found in Section 3 of the article.*

Other Comments: - Abstract: The reviewer does not necessarily agree that a separate tangential correction is necessary. A model rooted in the driving flow physics should be as easy and elegant as possible.

- *We agree. That is why we present a modification to the tip loss correction that does not need an additional parameter. The modification takes into account the rotation of the resulting force vector due to the induced velocity. This rotation leads to induced drag which results in a stronger tip loss effect in the in-plane direction compared to the out-of-plane direction.*

- Introduction (page 1): Not all AL models require a tip correction. This does not mean that using a tip correction is incorrect, but it should be acknowledged that there are other approaches.

- *See our comment to the second major concern above.*

- Introduction (page 2): The review of recent literature is incomplete. Some work has been published on the de-cambering effect and using free-wake method results as a look-up table for improved tip corrections. This should be acknowledged and it should be clarified what similar or other physics the proposed model captures.

- *We will extend the literature review to include this recent work.*

- Introduction (page 2, last sentence): Why only AD computations ? Inconsistent with manuscript title. Also, the comparisons are meaningless in the absence of experimental data and blade-resolved CFD simulations.

- *See our comments to the first and second major concern above.*

- Pages 2 and 3: Not sure if it is necessary to repeat descriptions of the models by Shen et al. and Wimshurst and Willden. It would make sense having the proposed model being described in conjunction with the older models (and not later on page 6)

- *We do not repeat model descriptions by Wimshurst and Willden in the article. The proposed model is found on page 3, immediately after the description of the model by Shen et al.*

- Table 1: Inadequate choice of test cases, see major comments above.

- *See our comments to the first major concern above.*

- Page 6, section 4.4: What is the physics behind 'h' ?

- *See our comments to the third major concern above.*

- Figure 3: There need to be available measured or CFD data for comparisons. How do differences between tip models integrate to deltas in thrust and power ?

- *See our comments to the first major concern above on the need for CFD and measurement data.*

- *We chose not to include the differences on integral thrust and power. Comparing these differences can be misleading because the load disctributions along the blades also differ between models away from the tip. Therefore looking at the integral thrust and power is more prone to error canceling than focusing on the detailed load distribution at the tip.*

- Page 8: Do not start sentence with 'Tab.' or 'Fig.'. More examples throughout the manuscript.

- *We will modify these sentences in the revised version.*

- Page 9, section 5.2: Justify why these model variations have been chosen. What is the effect on integrated thrust and power ?

- *We will explain the background for these model variations in more detail in the revised manuscript. We will remove the last line in the plots ($F_1^{Test}$: $F_n$ + $F_t$ fit:)where we fit the axial and tangential forces independently, because it is confusing and not necessary for the conclusions of the article. The updated descriptions are:*

  $F_1^{Shen}$: $F_n$ fit: The tip loss for $F_n$ and $F_t$ is computed using Eq. (2). The parameters are chosen to fit $F_n$. This corresponds to using Shen's tip loss correction

  $F_1^{Test}$: $F_n$ fit: The tip loss for $F_n$ and $F_t$ is computed using Eq. (7). The parameters are chosen to fit $F_n$. The purpose of this to obtain a better agreement of the axial force. This is possible because the 'test' function has an additional parameter.

  $F_1^{Shen}$: $F_n$ fit + $F_t$ mod: The tip loss for $F_n$ is computed using Eq. (2). $F_t$ is computed using Eq. (6). This means that the modified tip loss correction proposed here is used together with Shen's tip loss correction.

  $F_1^{Test}$: $F_n$ fit + $F_t$ mod: The tip loss for $F_n$ is computed using Eq. (7). $F_t$ is computed using Eq. (6). Thus the modified tip loss correction is based on a more closely matching axial force. Based on this the quality of the tangential load correction can be investigated with less error progression from the axial load correction.

- *See our comment above on integral thrust and power.*

- Figure 5: What is 'Reference' ? Add HAWC2 BEM to legend so this is not confusing. This figure (as are others) is of no use as there are no data for comparison. In particular, one would hope for improved comparisons against data for the proposed new model and tangential forces.

- *We will call it 'Reference: HAWC2 BEM' to make it less confusing.*

- *See our comments to the first major concern above on the need for CFD and measurement data.*

- Page 11, Conclusions: The second paragraph is unclear and not precise, e.g. ". . . when the fit for the axial loading is good".

- *We will clarify the conclusions. This sentence was meant to illustrate that the present modification for the tangential loading will only work if there is a good agreement on the axial loading.*

- Page 11, Conclusions: The last sentence of the manuscript is very concerning as it states that ". . . refitting is not required if a good fit of the normal force could be obtained using the original tip loss model" How can one know what to choose a-priori, if the method is used as a predictive tool ?

- *It is definitely feasible to run a BEM computation before an actuator disc computation in order to tune the tip loss factor. The pure BEM computation is orders of magnitude faster than the AD method.*